# BM2CP: Efficient Collaborative Perception with LiDAR-Camera Modalities

**Binyu Zhao, Wei Zhang\*, Zhaonian Zou**
School of Computer Science and Technology
Harbin Institute of Technology, China
byzhao@stu.hit.edu.cn, {weizhang,znzou}@hit.edu.cn

**Abstract:** Collaborative perception enables agents to share complementary perceptual information with nearby agents. This would improve the perception performance and alleviate the issues of single-view perception, such as occlusion and sparsity. Most existing approaches mainly focus on single modality (especially LiDAR), and not fully exploit the superiority of multi-modal perception. We propose a collaborative perception paradigm, *BM2CP*, which employs LiDAR and camera to achieve efficient multi-modal perception. It utilizes LiDAR-guided modal fusion, cooperative depth generation and modality-guided intermediate fusion to acquire deep interactions among modalities of different agents, Moreover, it is capable to cope with the special case where one of the sensors, same or different type, of any agent is missing. Extensive experiments validate that our approach outperforms the state-of-the-art methods with $50\times$ lower communication volumes in both simulated and real-world autonomous driving scenarios. Our code is available at https://github.com/byzhaoAI/BM2CP.

**Keywords:** Multi-Agent Perception, Multi-Modal Fusion, Vehicle-to-Everything (V2X) Application

## 1 Introduction

Collaborative perception enables agents to share complementary perceptual information with their nearby agents. This would fundamentally alleviate the issues of single-agent perception, such as occlusion and sparsity in raw observations. Recently, different strategies have been proposed to implement collaborative perception. These approaches can be divided into three categories: LiDAR based collaboration [1, 2, 3, 4, 5, 6, 7, 8, 9, 10, 11, 12, 13, 14, 15], camera based collaboration [16, 17, 18, 19] and multi-modal based collaboration [20].

Intuitively, different types of sensors can provide heterogeneous perceptual information at different levels, and thus, more accurate perception could be achieved through multi-modal analysis. However, most existing approaches are not multi-modal based methods, and present better performance only using LiDAR. Take camera and LiDAR as example, fusing the two modalities straightforwardly will bring negative impacts on perception performance, which is demonstrated by experiments in Table 3a. Camera captures rich semantics and contexts in a fixed view, but lacks the information of distance. Thus distance information, i.e. depth, will be estimated at first generally, which could help lift the camera representations from 2D to 3D to align with LiDAR representations. But the estimation brings uncertainty and have negative effect to modal fusion and subsequent collaborative feature fusion.

Therefore, it poses challenges to build a well-behaved collaborative perception method with LiDAR-camera modalities: (a) How to generate depth information? (b) How to fuse the LiDAR data and camera data effectively? (c) How to collaborate between agents with multi-modal data?

7th Conference on Robot Learning (CoRL 2023), Atlanta, USA.

In this paper, we answer the aforementioned questions by proposing a Biased Multi-Modal Collaborative Perception (*BM2CP*) method including three components: (a) cooperative depth generation. A hybrid strategy is applied to provide more reliable depth distribution, which combines projection from ego and nearby LiDARs and prediction from ego camera; (b) biased multi-modal fusion. A preferable modal fusion is obtained by LiDAR-guided feature selection, which approve the importance of LiDAR representations and utilize it to select useful camera representations; (c) modality-guided collaborative fusion. A preference threshold mask is generated to filter bird's-eye-view (BEV) features, which achieves multi-view multi-modal critical feature sharing. Meanwhile, a flexible workflow makes *BM2CP* capable to deal with the case that one of the sensors, which is same or different type, of any agent is missing.

In summary, our main contributions are threefold:

- We propose a novel framework for multi-modal collaborative perception, where modal fusion is guided by LiDAR and collaborative fusion is guided by modality. Besides, it can handle the case where modality data is incomplete.

- We design LiDAR-guided depth generation and biased modal fusion, which achieves deeper interactions between LiDAR and camera modalities. The message containing information of depths and features is exchanged among agents, which achieves better modal feature learning and efficient communication. These designs encourage sufficient feature fusion in both modality aspect and collaboration aspect.

- Extensive experimental results and ablation studies on both simulated and real-world datasets demonstrate the performance and efficiency of *BM2CP*. It achieves superior performances with 83.72%/63.17% and 64.03%/48.99% AP at IoU0.5/0.7 on OPV2V [9] and DAIR-V2X [21]. When all camera sensors are missing, the performance is still comparable to other state-of-the-art LiDAR based methods.

## 2 Related Works

**Collaborative perception.** As aforementioned stated, Most collaborative methods are LiDAR based, focusing on different issues such as performance [1, 3, 6, 8, 12], bandwidth trade-off [11, 18], communication interruption [7], latency [13] and pose error correction [5, 15]. For camera based methods, Xu et al. [18] propose the first attention-based multi-view cooperative framework, which shares features in BEV with Transformer [22]. Hu et al. [19] first conduct depth estimation and share it to reduce the impact of erroneous depth estimates. However, the depth ground truth is required to be collected in advance, which limits the generalization of the method. For LiDAR-camera based methods, Zhang et al. [20] use limited number of image pixels, which are in 2D predicted bounding box, to generate virtual 3D point clouds. Whereas the modality for collaboration is still LiDAR. Xiang et al. [23] focus on constructing attention-based collaborative network. The method requires that different agents provide heterogeneous modalities.

**Absence of sensor data.** To the best of our knowledge, none of multi-agent collaborative perception approach addresses the absence of sensor data issue. Recently, Li et al. [24] propose a solution with RADAR and LiDAR for single-agent perception that fills the missing sensor data with zero value, and uses the teacher-student model with exponential moving average (EMA) to learn equally from both modalities.

*BM2CP* constructs a flexible and straightforward workflow to overcome the absence of sensor data with available modalities. The proposed networks do not need any fine-tuning. Meanwhile, it is worth emphasizing that camera cannot accomplish the perception task independently, especially when each agent is only equipped with one camera. In this situation, reliable depth generation would become impossible and inferior perceptual information might be produced. This can be inferred from the research of Hu et al. [19] as well. Their experiments on DAIR-V2X dataset [21] show that the perception generated through combining cameras and depth ground truth is much worse than that generated based on LiDAR data.

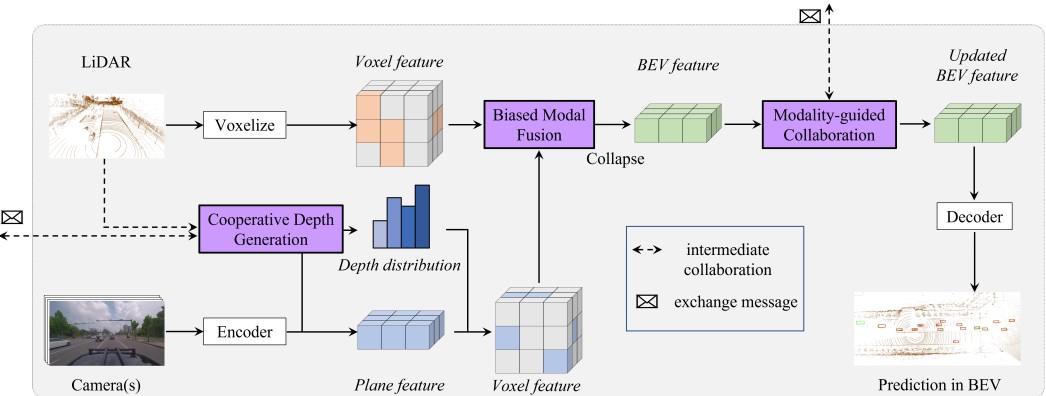

Figure 1: The framework of *BM2CP*. Colors indicate different modal voxels: orange for camera, which is obtained only from camera; blue for LiDAR, which is obtained only from LiDAR; green for LiDAR-camera hybrid, which is obtained from both modalities; and gray for normal, which receives no modal information from each modality and is filled with 0. In the object detection task example, green boxes are ground truths and red boxes are detected vehicles.

## 3 Efficient Collaborative Perception with LiDAR-Camera Modalities

Our designs include (a) *cooperative depth generation*, which is guided by LiDAR and shares depth information among agents; (b) *biased multi-modal fusion*, which achieves sufficient local feature learning; and (c) *modality-guided collaborative fusion*, which shares critical BEV detection features to improve representation and performance. The overall framework is illustrated in Fig 1.

### 3.1 Problem formulation

To the best of our knowledge, there does not exist any problem formulation about multi-agent collaborative perception with LiDAR-camera modalities. Here, we try to provide a feasible definition of LiDAR and camera perception fusion in voxel space. Considering $N$ agents in the scenario, let $I_i \in \mathcal{R}^{W \times H}$ be the RGB image collected by the camera of the $i$-th agent, where $W$ and $H$ are the height and width of image. Let $P_i \in \mathcal{R}^{N \times 3}$ be the point cloud collected by LiDAR of the $i$-th agent, where $N$ is the number of point clouds. $Y_i$ is the corresponding ground truth of detection. Agents exchange their positions and relative poses to build a communication graph.

In order to fuse modal features in voxel space, the point cloud and the image need to be voxelized.

$$\mathbf{V}_i^l = f_{voxelize}(P_i), \mathbf{V}_i^c = f_{img\_ext}(I_i) \otimes f_{dep\_est}(I_i) \tag{1}$$

where $f_{voxelize}(\cdot)$ is a series of operations to obtain voxel features from point cloud, which are similar to the operation defines in Lang et al. [25]. $f_{img\_ext}(\cdot)$ and $f_{dep\_est}(\cdot)$ are the feature extractor and depth estimator based on raw image. Plane features can be obtained through $f_{img\_ext}(\cdot)$. The $\otimes$ operation produces the voxel features, which expands the plane features a new dimension in Z-axis.

Then, we fuse point cloud features $\mathbf{V}_i^l \in \mathcal{R}^{X \times Y \times Z}$ and image features $\mathbf{V}_i^c \in \mathcal{R}^{X \times Y \times Z}$ by cell and collapse the fused voxel features $\mathbf{V}_i^f \in \mathcal{R}^{X \times Y \times Z}$ to BEV.

$$\mathbf{V}_i^f = f_{modal\_fuse}(\mathbf{V}_i^l, \mathbf{V}_i^c), \mathbf{F}_i = f_{collap}(\mathbf{V}_i^f) \tag{2}$$

where $f_{modal\_fuse}(\cdot, \cdot)$ and $f_{collap}(\cdot)$ denote the modality fuse operation and collapse operation, respectively. Collapse operation integrates the dimension of Z-axis to the channel dimension [25]. After collapsing, BEV feature $\mathbf{F}_i \in \mathcal{R}^{X \times Y}$ will be packed and transmitted to nearby agents based on communication graph.

Each agent aggregates the received features with its own BEV feature and finally conduct prediction for a specific task, such as object detection or scenario segmentation.

$$\tilde{\mathbf{F}}_i = f_{feat\_fuse}(\mathbf{F}_i, \{\mathbf{F}_{j\rightarrow i}\}_{j\in N_i}), \hat{Y}_i = f_{decoder}(\tilde{\mathbf{F}}_i) \tag{3}$$

where $\tilde{\mathbf{F}}_i$ is the aggregated feature. $\mathbf{F}_{j\rightarrow i}$ is the warped feature based on relative poses of the $j$-th agent and the $i$-th agent. $N_i$ is the nearby agent set that the $i$-th agent can communicate with. $f_{feat\_fuse}(\cdot)$ denotes the feature fuse operation and $f_{decoder}$ denotes the decoder for prediction.

The objective of *BM2CP* is to minimize the distance between predicted and the ground truth detection $\sum_i g(\hat{Y}_i, Y_i)$, where $g(\cdot, \cdot)$ is the evaluation metric.

### 3.2 Cooperative depth generation

A hybrid strategy, *prediction&projection*, is applied to reduce the erroneous of estimation and obtain a reliable depth distribution. *Prediction* is used to predict pixel-wise depth using convolutional blocks. In order to reduce the number of model parameters, image feature extractor $f_{img\_ext}(\cdot)$ and depth estimator $f_{dep\_est}(\cdot)$ utilize a shared encoder and independent heads, i.e. depth head and image head. Each head is composed of several convolutional layers and training from scratch. Similar to the method in Reading et al. [26], the predicted depths are classified to a series of discrete values. The number of classes is the same as that of discretized depth bins. *Projection* is applied to transform LiDAR point clouds to RGB image coordinates. Let $P_i = \{(x_i, y_i, z_i, 1)\}$ be the homogeneous coordinates of point cloud, where $(x_i, y_i, z_i)$ is the 3D coordinate. Let $T_{ldr2cam}$ be the mapping from LiDAR sensor to camera sensor, and $T_{cam2img}$ represent the mapping from camera sensor to the RGB image. The overall mapping from LiDAR to RGB image is

$$I_i^{'} = T_{ldr2img}P_i = T_{ldr2cam}T_{cam2img}P_i \tag{4}$$

where $I_i^{'} = \{(u_i, v_i, d_i)\}$ is the projected image with depth information, and $(u_i, v_i)$ is the 2D coordinate, $d_i$ is the corresponding depth of each pixel. $T_{ldr2cam}$ and $T_{cam2img}$ are equal to the extrinsic and intrinsic matrices of camera, respectively. The projected depths are also mapped to discrete values corresponding to the discretized depth bins.

Considering that some pixels may not have any projection depth, while some pixels have multiple depths to map, the hybrid strategy is implemented as follows: a) For pixel with no projection depth, it obtains the depth through *prediction* strategy; b) For pixel with only one projection depth, no extra operation is required; c) For pixel with multiple projection depths, it selects the minimum depth. According to the principle of imaging, each pixel only presents the attribute of nearest object in reflected lights while the other objects are occluded. Therefore, it is more reasonable to select the minimized depth which is the closest to camera.

On the other hand, point clouds from nearby agents contain different depth information. Intuitively, the 3D location of a correct depth candidate is spatially consistent through viewpoints of multiple agents. Therefore, more reliable depth distribution could be obtained through communications. Since the depths projected from ego agent is more accurate, the depths from nearby agents are only used to replace predicted depths.

### 3.3 Biased multi-modal fusion

Motivated by the fact that LiDAR-based detectors usually surpass camera-based counterparts, we take LiDAR as the guiding modality to achieve multi-modal fusion and generate fused voxel features $\mathbf{V}_i^f$. The illustration is presented in Fig 2a. The voxels are grouped into four categories: LiDAR voxels, camera voxels, LiDAR-camera hybrid voxels, and normal voxels. LiDAR voxels and camera voxels are features only obtained from LiDAR branch and camera branch, respectively. Hybrid voxels are features obtained from both branches. Normal voxels denote it receives no modal information from each modalities, which are filled with 0.

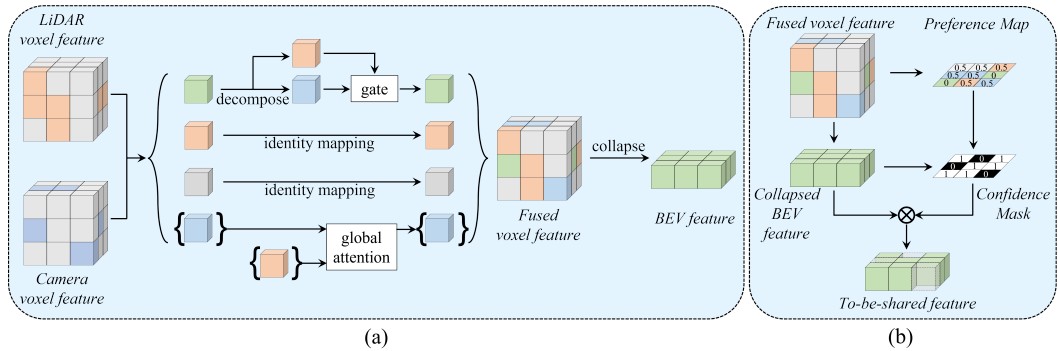

Figure 2: (a) LiDAR-guided modal fusion. (b) Modality-guided preference map and confidence mask generation. Colors indicate different modal voxels: orange for camera, blue for LiDAR, green for LiDAR-camera hybrid, and gray for normal.

For LiDAR-camera hybrid voxels, the fusion result $\tilde{v}_H$ is conditioned by LiDAR, which is formulated as $\tilde{v}_H = Conv([ReLU(Conv(v_L)) * v_C, v_L])$, where $v_L$ and $v_C$ denote cells from the same position of LiDAR and camera, respectively. $[\cdot, \cdot]$ denotes concatenation.

For camera only voxels, LiDAR information is not contained in the same cell. Therefore, we conduct global attention to filter camera voxel features $\mathbf{V}^c$. The guidance matrix $A_{x,y,z}$ comes from overall LiDAR voxel features $\mathbf{V}^l$

$$A_{x,y,z} = \begin{cases} 0 & MHA(\mathbf{V}^l, \mathbf{V}^c, \mathbf{V}^c) < threshold \\ 1 & MHA(\mathbf{V}^l, \mathbf{V}^c, \mathbf{V}^c) > threshold \end{cases} \tag{5}$$

where $MHA(\cdot, \cdot, \cdot)$ is the multi-head attention [22] which outputs the scaled dot-product attention weight. The threshold is empirically set as 0.5. The final camera voxel sets $\{\tilde{v}_C\}$ is formulated as $\{\tilde{v}_C\} = A_{x,y,z} \times \mathbf{V}^c_{x,y,z}$, which collects the features that $A_{x,y,z}$ is not 0.

Since LiDAR voxels and normal voxels are not affected by camera voxel features, their fusion results can be acquired through identity mapping. Finally, biased modal fusion is achieved and the fine-grained voxel features $\mathbf{V}^f_i$ are collapsed to BEV feature.

### 3.4 Multi-agent collaborative perception

We design a modality-guided collaboration to select the most critical spatial features and promote efficient communication, as Fig 2b illustrated.

First, a plane preference map $T \in \mathcal{R}^{X \times Y}$ is generated based on the fused voxel features $\mathbf{V}^f_i \in \mathcal{R}^{X \times Y \times Z}$. A threshold is assigned to each cell $T_{m,n}$ of the preference map based on a set of voxels that can be collapsed to it in Z-axis. From the set of voxels, we select a preferred voxel to decide the threshold of corresponding cell. The preferred order is *hybrid >LiDAR >camera >normal*. Examples and more cases can be found at Appendix A.3. Since the hybrid voxels is guided by LiDAR and contains sufficient multi-modal information, the hybrid threshold is set as 0. For a better performance-bandwidth trade-off, the threshold for other types of cell is set as 0.5.

Then, a binary confidence mask $M \in \mathcal{R}^{X \times Y}$ is generated based on the preference map $T$ and the collapsed BEV feature $\mathbf{F}_i \in \mathcal{R}^{X \times Y}$. First, a classification head $f_{cls}(\cdot)$ is used to evaluate the importance of each cell in BEV feature, and $f_{cls}(\mathbf{F}_i) \in [0, 1]$. On the other hand, preference map provides a threshold $T_{m,n}$ for each cell in BEV feature at the same location $(m, n)$. Then we compare the value of importance at the position $(m, n)$ in $f_{cls}(\mathbf{F}_i)$ and the threshold $T_{m,n}$. When the value of importance is greater than threshold, $M_{m,n} = 1$ and the cell at the same location $(m, n)$ in BEV feature $\mathbf{F}_i$ is regarded as critical, and will be broadcast to nearby agents. When the value of importance is smaller, $M_{m,n} = 0$.

Table 1: 3D detection quantitative results on OPV2V dataset and DAIR-V2X dataset. *Comm* is the communication volumes in log-scale.

| Dataset | OPV2V | | | DAIR-V2X | | |
|---|---|---|---|---|---|---|
| Method | Comm | AP@0.5 | AP@0.7 | Comm | AP@0.5 | AP@0.7 |
| No Fusion | 0 | 61.65 | 43.26 | 0 | 52.75 | 45.63 |
| Late Fusion | 18.43 | 74.27 | 57.45 | 18.62 | 53.88 | 38.12 |
| When2com (CVPR'20) | 20.17 | 69.12 | 53.76 | 20.31 | 51.88 | 37.05 |
| V2VNet (ECCV'20) | 22.56 | 78.54 | 59.42 | 22.90 | 58.80 | 43.75 |
| DiscoNet (NeurIPS'21) | 21.61 | 77.01 | 58.50 | 21.85 | 55.17 | 39.84 |
| CoBEVT (CoRL'22) | 21.07 | 81.37 | 61.32 | 21.31 | 50.70 | 39.39 |
| V2X-ViT (ECCV'22) | 20.21 | 80.92 | 61.23 | 20.45 | 56.75 | 39.90 |
| Where2comm (NeurIPS'22) | 15.64 | 79.67 | 60.15 | 16.54 | 60.85 | 46.48 |
| BM2CP | 11.13 | 83.72 | 63.17 | 11.01 | 64.03 | 48.99 |

After collaboration, multi-head attention $MHA(q, k, v)$ is implemented to aggregate these critical BEV features from agents and generates updated BEV feature $\tilde{\mathbf{F}}_i$, where $q = k = v = [\mathbf{F}_i, \{\mathbf{F}_{j \to i}\}_{j \in N_i}]$. $[\cdot, \cdot]$ is concatenation operation and $\{\cdot\}$ denotes the feature set from nearby agent set $N_i$. The updated BEV feature is finally used to predict the detection with task-specific decoder.

### 3.5 Robustness against missing sensor

*BM2CP* is capable to deal with the case when one of the sensors is missing through a flexible and straightforward workflow. Suppose that the camera sensor is now absent and RGB image is not available. In modal fusion step, LiDAR voxels and normal voxels remain, and they are used as the fusion results through identity mapping. In collaborative fusion step, since no hybrid voxels exist, the threshold of cells is uniformly set as 0.5. The paradigm is similar when LiDAR sensor is missing. By conducting this, *BM2CP* adapts well to the modality-unavailable cases. More discussion about missing sensor can be found at Appendix A.4. The experimental results are collected in Sec 4.2.

## 4 Experimental Results

### 4.1 Experimental setup

**Dataset.** We conduct experiments of 3D object detection task on OPV2V dataset [9] and DAIR-V2X dataset [21]. OPV2V dataset is a vehicle-to-vehicle (V2V) collaborative perception dataset, which is co-simulated by OpenCDA and Carla [27]. The perception range is $40m \times 40m$. DAIR-V2X dataset is the first public real-world collaborative perception dataset. Each sample contains a vehicle and an infrastructure, and they are equipped with a LiDAR and a front-view camera. The perception range is $201.6m \times 80m$.

**Compared methods.** We consider comparisons with following LiDAR-based methods: *No Fusion*, *Late fusion*, *When2com* [4], *V2VNet* [3], *DiscoNet* [18], *V2X-ViT* [12] and *Where2comm* [11]. Among these methods, *No Fusion* is considered as the baseline which only uses individual observation. *Late fusion* shares the detected 3D bounding boxes with nearby agents. Rest are state-of-the-art (SOTA) LiDAR-based collaborative perception algorithms.

**Implementation details.** We re-implement all methods based on the pyTorch [28] framework and OpenCOOD [9] codebase with Adam [29] optimizer and multi-step learning rate scheduler. In order to compare communication volumes fairly, all compared methods use the same architectures and follows PointPillar [25] with no feature compression. Weighted cross entropy loss is used. The detection results are evaluated by Average Precision (AP) at Intersection-over-Union (IoU) threshold of 0.50 and 0.70.

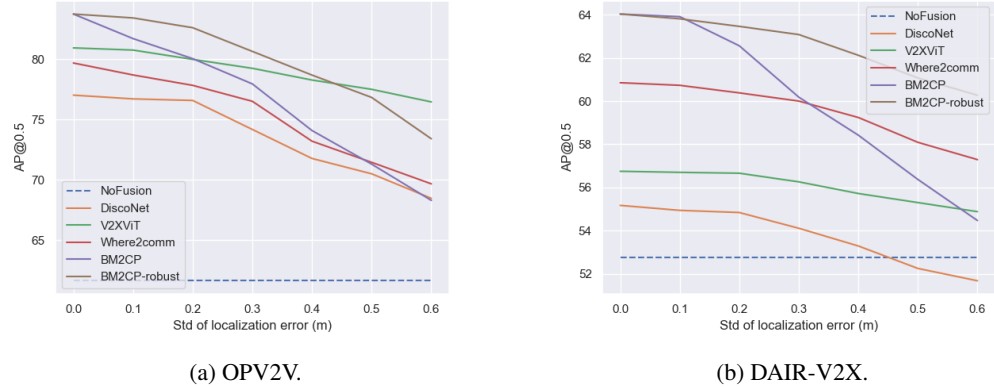

(a) OPV2V.

(b) DAIR-V2X.

Figure 3: Robustness to localization error. Gaussian noise with 0 mean and varying std is introduced.

Table 2: Evaluation results against missing sensor on DAIR-V2X dataset. *Camera* or *LiDAR* denotes the sensor that the agent is equipped with.

| Ego agent | Nearby agent(s) | AP@0.5 | AP@0.7 |
|-----------|-----------------|--------|--------|
| Camera | LiDAR | 19.21 | 5.74 |
| Camera | Both | 20.41 | 6.05 |
| LiDAR | LiDAR | 61.59 | 47.30 |
| LiDAR | Both | 62.71 | 47.64 |
| Both | Camera | 53.31 | 43.86 |
| Both | LiDAR | 63.54 | 48.38 |
| Both | Both | 64.03 | 48.99 |

## 4.2 Quantitative Evaluation

**Comparison with baseline and SOTA methods.** Table 1 shows the comparisons with recent methods in terms of communication volumes and detection performance. Mathematically, the communication volume is calculated in log-scale by $log_2(|\mathbf{F}_{i \to j}|)$, where $|\cdot|$ is the $L0$ norm which counts the non-zero elements in BEV features. It is observed that *BM2CP*: (a) achieves a superior perception-communication trade-off; (b) achieves significant improvements over previous SOTA methods on both datasets; (c) achieves a better performance with extremely less communication volume, which is 105 times less than baseline and 46 times less than *Where2comm*.

**Robustness against the absence of modality.** Table 2 shows the results with the setting of missing sensor. It is observed that: (a) the performance degrades severely when one of the agents is only equipped with camera sensor. This is consistent with the conclusion that the performance is poor when only camera sensor works; (b) The performance (61.59% on AP@IoU0.5 and 47.30% on AP@IoU0.7) is comparable with SOTA methods, when agents only use LiDAR for perception; (c) The performance drop slightly when one of the agents miss the camera sensor.

**Robustness to localization noise.** We also evaluate the robustness to localization noise following the setting in *V2VNet* [3] and *V2X-ViT* [12]. Gaussian noise with a mean of $0m$ and a standard deviation of $0m - 0.6m$ is used, and the results are shown in Fig 3. Unfortunately, performance degrading happens when localization noise increases. The reason comes from the *cooperative depth generation*. The shared depth distributions are broadcast based on the localization and relative pose of each agent. Therefore, it intensifies the errors and provides wrong depth information for images. To solve this problem, we correct relative pose before depth and feature communication and name it *BM2CP-robust*. Detailed process can be found at Appendix A.5. Comparing the results of *BM2CP-robust* with *BM2CP*, the performance degrading gets alleviated evidently.

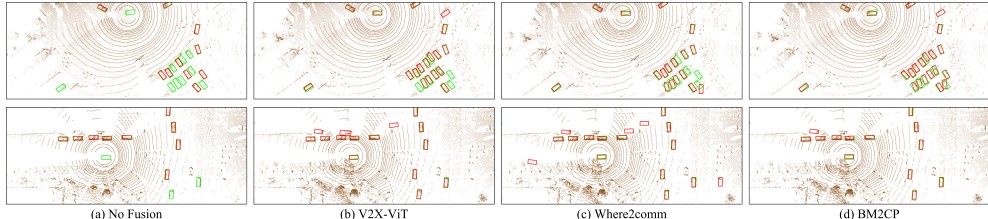

| (a) No Fusion | (b) V2X-ViT | (c) Where2comm | (d) BM2CP |

Figure 4: Qualitative results on DAIR-V2X dataset. Ground truths are colored in green and predictions are colored in red. *BM2CP* detects more objects than *V2X-ViT* and *Where2comm* in the first row, and detects less false positive objects in the second row.

Table 3: Component ablation studies on DAIR-V2X dataset.

| (a) Modal fusion strategy. | | (b) Depth projection. | | (c) Collaborative strategy. | |
|---|---|---|---|---|---|
| Strategy | AP@0.5/0.7 | Projection | AP@0.5/0.7 | Strategy | AP@0.5/0.7 |
| No | 50.17/37.33 | No | 52.84/38.07 | Max | 61.33/46.02 |
| Equal | 58.33/42.13 | Ego | 60.31/44.42 | Concat | 62.25/46.46 |
| Biased | 60.31/44.42 | All | 62.25/46.46 | Attention | 64.03/48.99 |

### 4.3 Qualitative Evaluation

Fig 4 shows the comparison with *No Fusion*, *V2X-ViT* and *Where2comm*. *BM2CP* achieves more complete and less false negative detection. The reason is that *BM2CP* leverages multi-modal feature fusion in critical voxels and employs modality-guided cell-level confidence mask to achieve more comprehensive fusion. We also visualize the projected depths, which can be found at Appendix B.3.

### 4.4 Ablation Study

Ablation study is conducted to investigate the effectiveness of the main components in our method. The results are presented in Table 3. We also conduct ablation studies on the number of agents cameras. They can be found at Appendix B.4.

**LiDAR-guided modal fusion.** As shown in Table 3a, equally multi-modal fusion can result in a minor drop in performance. It proves the importance of the guidance by LiDAR.

**LiDAR-based depth generation.** We investigate the effect of using projected depths from ego and nearbys. Results in Table 3b indicate LiDAR-based cooperative depth generation is essential.

**Modality-guided collaborative fusion.** We compare our masking-attention strategy with strategies of max fusion and concatenate fusion. Table 3c show that both max fusion and concatenate fusion lead to performance degradation and increase the demand of bandwidth.

## 5 Conclusion

We propose a novel framework, termed *BM2CP*, for multi-agent collaborative perception, which focuses on fusion with LiDAR-camera modalities. We adopt LiDAR-guided modal fusion to achieve reasonable and comprehensive modal feature learning, apply cooperative depth generation to enhance the modal fusion with more reliable depth information, and propose modality-guided collaborative fusion for more efficient and critical feature fusion. Extensive experiments demonstrate the superior performance and the effectiveness of designed components.

**Limitation and Future Works.** Although *BM2CP-robust* significantly alleviates performance degrading, it increases the runtime during training and test. Further efforts is needed to reduce the complexity of overall workflow and computation. Besides, validations on more datasets are critical to prove its generalization.

**Acknowledgments**

We would like to thank all the reviewers for their helpful and valuable comments.

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

# Appendix

## A  Method

### A.1  Motivation

Our motivation comes from two aspects. Firstly, depth is the major gap for RGB image to lift up to voxels, but part of correct depths could be acquired from LiDARs of an agent or nearby agents. Thus, depth information could be transmitted among modalities of agents. This allows compensation from LiDAR in different views, and reduces the uncertainty of the infinite depth prediction. Secondly, the transmitted data should include detection clues to provide refined and complementary information, which can fundamentally overcome inevitable limitations detected by single modality or single agent.

### A.2  Coordinate transformation among LiDAR, camera and RGB image

If the LiDAR coordinate system is regarded as the world coordinate system, the 3D coordinate of point $W$ could be:

$$W_{world} = \begin{bmatrix} x_{world} \\ y_{world} \\ z_{world} \end{bmatrix} \tag{6}$$

We also have the camera coordinate and image coordinate of point $W$ as

$$W_{cam} = \begin{bmatrix} x_{cam} \\ y_{cam} \\ z_{cam} \end{bmatrix}, W_{img} = \begin{bmatrix} x_{img} \\ y_{img} \end{bmatrix} \tag{7}$$

Its world homogeneous coordinate in camera coordinate system and its camera homogeneous coordinate in image coordinate system are

$$W_{world\_h} = \begin{bmatrix} x_{world} \\ y_{world} \\ z_{world} \\ 1 \end{bmatrix}, W_{img\_h} = \begin{bmatrix} x_{img} \\ y_{img} \\ 1 \end{bmatrix} \tag{8}$$

Suppose that $E$ is the transformation matrix from LiDAR coordinate system to camera coordinate system and $I$ is the transformation matrix from camera coordinate system to image coordinate system, The inverse matrix of $E$ and matrix $I$ are

$$E_{4\times4}^{-1} = \begin{bmatrix} r_{x1} & r_{y1} & r_{z1} & t_x \\ r_{x2} & r_{y2} & r_{z2} & t_y \\ r_{x3} & r_{y3} & r_{z3} & t_z \\ 0 & 0 & 0 & 1 \end{bmatrix}, I_{3\times3} = \begin{bmatrix} f_x & 0 & u \\ 0 & f_y & v \\ 0 & 0 & 1 \end{bmatrix} \tag{9}$$

Generally, $E$ and $I$ are the extrinsic matrix and intrinsic matrix of camera, which are given by dataset [21].

Then we have

$$W_{cam\_h} = E_{4\times4} * W_{world\_h}, W_{img} = \frac{1}{z_{cam}} * I_{3\times3} * W_{cam} \tag{10}$$

### A.3  Generate preference map

For example, when the voxel set is composed of $\{v_{hybrid}, v_{camera}, v_{normal}\}$, the $v_{hybrid}$ is the preferred voxel and the threshold of corresponding cell in preference map is assigned as hybrid threshold; When the voxel set is composed of $\{v_{LiDAR}, v_{normal}^1, v_{normal}^2\}$, the $v_{LiDAR}$ is the preferred voxel and the threshold of corresponding cell in preference map is assigned as LiDAR threshold.

We also visualize four typical cases when generating one cell of preference map, which are shown in Fig 5.

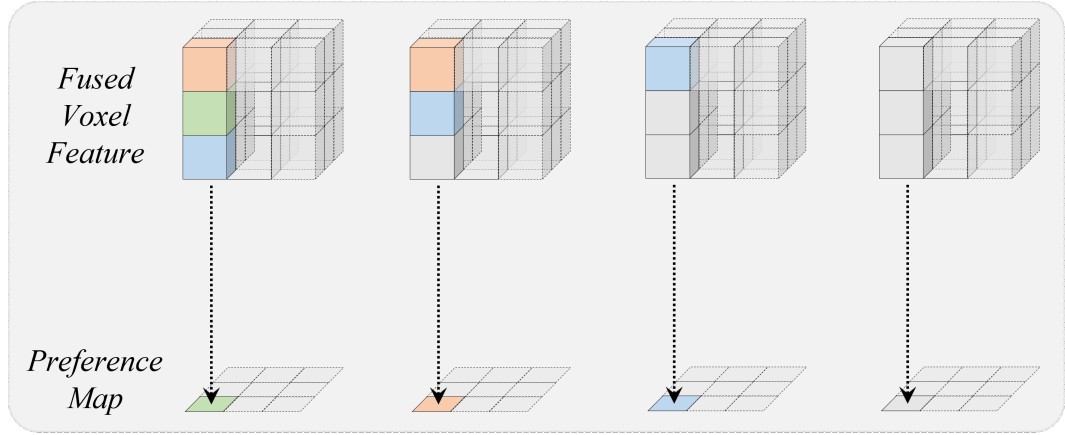

Figure 5: Four typical cases when generating one cell of preference map.

## A.4 Robustness against missing sensor

There are mainly 2 cases of missing sensor. Case 1 is shown in Fig 6a. Suppose that the camera sensor is now absent and RGB image is not available. In modal fusion step, LiDAR voxels and normal voxels remain, and they are used as the fusion results through identity mapping. In collaborative fusion step, since no hybrid voxels exist, the threshold of cells is uniformly set as 0.5. The paradigm is similar in case 2 when LiDAR sensor is missing. The modal fusion step is shown in Fig 6b. By conducting this, *BM2CP* adapts well to the modality-unavailable cases.

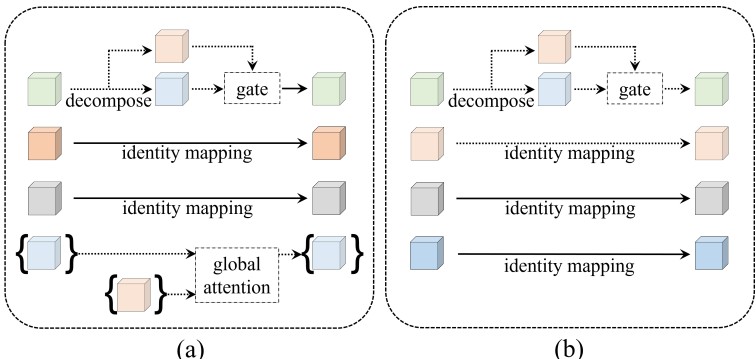

Figure 6: The workflow of biased multi-modal fusion when camera (a) or LiDAR (b) is missing. Existing LiDAR voxels and normal voxels in case (a), or camera voxels and normal voxels in case (b) will conduct identity mapping as the final fusion results. Colors indicate different modal voxels: orange for camera, blue for LiDAR, green for LiDAR-camera hybrid, and gray for normal. Transparent cells and dashed arrows indicate the corresponding type of voxel features do not exist.

## A.5 Robust BM2CP

*BM2CP-robust* corrects relative pose before depth and feature communication. Specifically, single-agent 3D object detection is first conducted to estimate local bounding boxes and their uncertainty for each agent with LiDAR voxel features. Then we conduct internal agent-object pose graph optimization [15] for each agent to correct relative pose $\xi_{j\to i}$, where $i$ and $j$ are ego agent and nearby agent, respectively. The corrected relative pose is used to correct shared depth maps and BEV features.

# B Experiments

## B.1 Detailed settings of architecture

We follow the default settings in OpenCOOD [9] codebase, which is also shown in Tab 4.

Table 4: Details of unified network architecture.

| Blocks | Settings |
|---|---|
| Voxel Feature Encoder (VFE) | use normalization and absolute 3D coordinates, 64 filters |
| PointPillar Scatter | 64-channel output |
| BEV backbone | ResNet backbone:
layers=$[3, 4, 5]$
strides=$[2, 2, 2]$
filters=$[64, 128, 256]$
upsample_strides=$[1, 2, 4]$
upsample_filters=$[128, 128, 128]$ |
| Shrink Header | shrink from 384 channels to 256 channels with stride 3 |
| Detect Head | 256-channel output with 2 anchors |

## B.2 Detailed settings of experiments

Table 5: Details of unified network architecture.

| Method | optimizer | lr schedule | initial lr |
|---|---|---|---|
| No Fusion | Adam | multistep | 1e-3 |
| Late Fusion | Adam | multistep | 1e-3 |
| When2com (CVPR'20) | Adam | multistep | 1e-3 |
| V2VNet (ECCV'20) | Adam | multistep | 1e-3 |
| DiscoNet (NeurIPS'21) | Adam | multistep | 2e-3 |
| CoBEVT (CoRL'22) | Adam | multistep | 2e-3 |
| V2X-ViT (ECCV'22) | Adam | multistep | 2e-3 |
| Where2comm (NeurIPS'22) | Adam | multistep | 2e-3 |
| BM2CP | Adam | multistep | 1e-3 |

## B.3 More Visualizations

**Visualization of projected depths.** Fig 7 shows how depth distribution is empowered by LiDAR in *projection* design. In the scene, objects show evident contour against the background with lighter coloring. And more depths from nearby agents will further fill the depth in empty (white pixel in image).

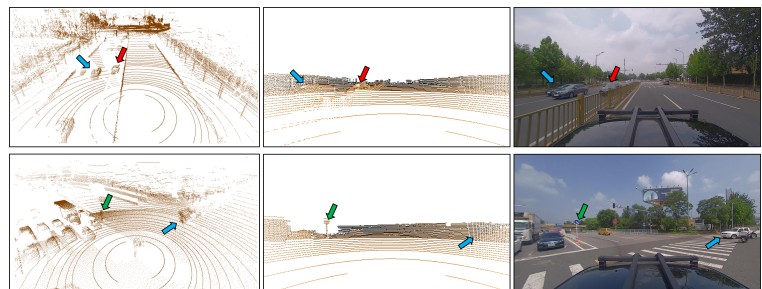

Figure 7: Two visualizations of projected depths from LiDAR coordinates to image coordinates. Paired arrows in colors indicate the same objects including car and sign in LiDAR data (left), camera data (right) and projected depth map (middle), respectively.

**More visualizations of comparison with SOTA methods.** Fig 8 shows more comparisons with *No Fusion*, *V2X-ViT* and *Where2comm*.

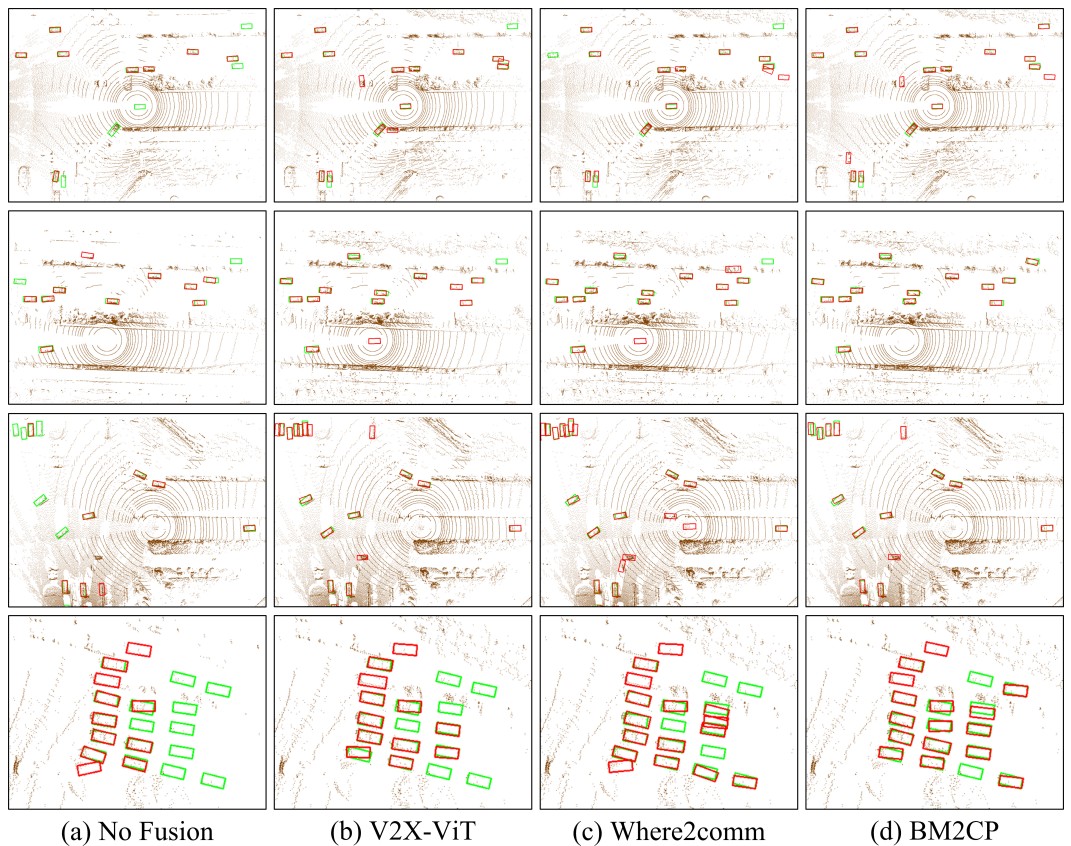

    (a) No Fusion           (b) V2X-ViT          (c) Where2comm        (d) BM2CP

Figure 8: More qualitative results in DAIR-V2X dataset. Ground truths are colored in green and predictions are colored in red.

## B.4  More Ablation Studies

**Number of agents.** We study the influence brought by the number of collaborative agents. As shown in Table 6a, increasing the number of collaborative agents can generally bring performance improvement on OPV2V dataset, whereas such gain will be more marginal when the number is greater than 4.

**Robustness to camera dropout.** We demonstrate the performance when the ego agent carries $n \in [1, 4]$ cameras in Table 6b. It can be seen that by the performance decreases but still maintain in an acceptable level.

Table 6: Ablation studies of the number of agents and cameras on OPV2V dataset.

(a) The number of agents.

| Number | AP@0.5 | AP@0.7 |
|--------|--------|--------|
| 1 | 58.05 | 28.78 |
| 2 | 67.83 | 34.66 |
| 3 | 75.73 | 48.57 |
| 4 | 79.29 | 57.85 |
| 5 | 83.72 | 63.17 |

(b) The number of cameras.

| Number | AP@0.5 | AP@0.7 |
|--------|--------|--------|
| 1 | 79.84 | 59.07 |
| 2 | 80.02 | 60.55 |
| 3 | 82.57 | 61.68 |
| 4 | 83.72 | 63.17 |

