# OpenReview forum: "BM2CP: Efficient Collaborative Perception with LiDAR-Camera Modalities"
_robot-learning.org/CoRL/2023/Conference — CoRL 2023 Poster_

### Official Review · Reviewer_oA8p · 2023-06-29

**Confidence:** 3
**Originality:** Fair
**Technical Quality:** Good
**Clarity Of Presentation:** Good
**Impact:** 3

**Recommendation:**

Weak Accept: I recommend accepting the paper, but will not argue for my recommendation if the majority of other reviewers have a different opinion.

**Review:**

The quality of the paper is okay. The fusion idea is a bit incremental, but I believe this framework might still be valuable in scenarios like autonomous driving and multi-robot collaboration tasks.

**Strengths**:
1. Strong detection results: the authors compare with two simple baselines and five SOTA approaches and consistently outperform other methods in the average precision measure.

**Weaknesses**:
1. Clarity: The problem formulation and methodology part is not very clear to me. Specifically, equation 1 provides little information: what is the $\otimes$ operator, and how is it formulated? What are normal voxels mentioned in line 139? In equation 7, what are the mappings $T_{lidar2cam}$ and $T_{cam2img}$, and why formulating $P_i$ like a 4d vector? My best guess is that those T are extrinsic/intrinsic matrices and P is like homogeneous coordinates, but the authors should make it clear in the paper. The categorical depth distribution in line 122 is not clear at first glance - I guess it might be related to the discretized depth bins mentioned in line 158, but again the authors should make it clear.
2. Generalizability: The authors only use one dataset (DAIR-V2X) throughout the paper. But there are also other datasets like OPV2V [1] and V2X-Sim [2], which have more agents (the authors mention that in DAIR-V2C only 2 agents are considered). Without conducting experiments on different datasets, it is hard to say whether this framework can be generalized to different datasets.
3. Ablation studies: there are many thresholds (line 147, line 186, line 188) and gate structures (ReLU, Sigmoid, Attention) in this framework. The authors should do experiments to provide the rationale they choose the one shown in the paper.

**References**

[1] Runsheng Xu, Hao Xiang, Xin Xia, Xu Han, Jinlong Liu, and Jiaqi Ma. OPV2V: An open benchmark
dataset and fusion pipeline for perception with vehicle-to-vehicle communication. ICRA, 2022.

[2] Yiming Li, Ziyan An, Zixun Wang, Yiqi Zhong, Siheng Chen, and Chen Feng. V2X-Sim: A virtual
collaborative perception dataset for autonomous driving. IEEE Robotics and Automation Letters, 7, 2022.

**Quality Of The Limitations Section:**

Limitations are addressed clearly

**Questions For Rebuttal:**

It will be great if the authors could address the questions/content from the **Weaknesses** section above. Besides:

1. What are the training time and the inference runtime for the proposed algorithm and the baselines? What is the hardware needed to run the algorithm?
2. In the "Cooperative depth generation" procedure, does the depth estimation network module use any pre-trained model, or entirely train from scratch?


**Robotics Focus:**

Highly relevant to robotics but no hardware experiments

**Summary Of Paper:**

This paper proposes a new fusion-based representation learning framework under the multi-agent and multi-modality (camera, LiDAR) setting. The fusion module takes LiDAR as input and decides which (camera/LiDAR) feature from which agent should be used. The authors conduct experiments on and compare with multiple strong baselines to show that their 3D object detection performance is better. They also show how their framework can handle the scenario where some sensors are missing.

**Summary Of Recommendation:**

I think the paper has a certain value in proposing a new framework for multi-agent multi-modal fusion for perception representation learning. Though they show strong results compared to many baselines, the paper is not written in a very clear way, the algorithm is only tested on one dataset and some important ablation studies are missing. Thus my decision is weak reject.

---

### Official Review · Reviewer_TJuY · 2023-07-14

**Confidence:** 3
**Originality:** Good
**Technical Quality:** Good
**Clarity Of Presentation:** Good
**Impact:** 3

**Recommendation:**

Weak Accept: I recommend accepting the paper, but will not argue for my recommendation if the majority of other reviewers have a different opinion.

**Review:**

The authors address an important aspect of the large-scale deployment of robots. They showcase a key advantage of leveraging sensor information across agents to extend the field of view and overcome sensor failures by bootstrapping measurements from nearby agents.

The general structure of the manuscript is adequate and manages to guide the reader through the approach by introducing the components step-by-step.

The biggest problem of the manuscript in its submitted form is that it suffers from several language (mostly grammar) errors that render certain sections difficult to read, e.g., the last paragraph in Sec. 2, and/or make it extremely hard to understand the technical method, e.g., Sec. 3.5.

In some sections, the manuscript fails to provide sufficient details to fully comprehend and evaluate the proposed method. Examples include the generation of depth prediction if no LiDAR data is available. The manuscript vaguely state "using neural network", more technical details would be appreciated.
Similarly, how are the cells of the voxel features collapsed along the z-axis? The classification into hybrid, LiDAR, etc. is clear but I do not understand which mathematical operation is used to aggregate the original voxels.

A major downside of the proposed method is the fact that it relies on ground truth poses. While this is mentioned in the limitations section, it should be communicated more clearly and earlier in the manuscript. In particular, as the authors state that the performance drastically drops if they add artificial noise to the poses. A more detailed ablation study including a quantitative analysis would help to better evaluate the performance of the method and whether publication at the current stage can be justified.

Similarly, the evaluation of the method is not very detailed given that the authors only use a single dataset. While it may be difficult to assess other datasets due to the limited availability of collaborative data, a fairly simple solution would be to leverage simulators such as Carla.

**Quality Of The Limitations Section:**

Additional details required

**Questions For Rebuttal:**

- Why is the minimum depth selected for pixels with more than one projected depth values? What is the intuition behind this, e.g., as opposed to averaging?
- In line 172, what is meant by "part pixels"?
- Is the confidence mask generated only based on the values of the preference map?
- What are the numbers mentioned in Sec 4.2.1 (b) based on? Taking the results from Table 1 does not yield improvements of 5.2% and 5.4% for IoU0.5 and IoU0.7, respectively.


**Robotics Focus:**

Highly relevant to robotics but no hardware experiments

**Summary Of Paper:**

The manuscript presents a new approach called BM2CP for multi-modal collaboration between multiple vehicles to address perception tasks in BEV space. The example given in this paper is 3D object detection.

The main contributions of this work are the following: First, a new LiDAR-camera fusion scheme to generate voxel features in BEV space, where the fusion is primarily guided by the LiDAR measurements. This fusion scheme is capable of handling the absence of either the LiDAR or the camera sensor. Second, a data communication approach to efficiently share information between agents in a local environment on a feature level.

The proposed approach is evaluated on a single dataset, namely DAIR-V2X, and demonstrates promising results given ground truth poses of the agents.

**Summary Of Recommendation:**

While I do appreciate the general idea of the submitted work, the manuscript in its current form requires major rewriting on a language level to clarify open questions. I would encourage the authors to address both of the listed limitations as they do not point to future research but are mostly related to a more in-depth evaluation of the developed method.

POST-REBUTTAL:
The authors extensively replied to all of my questions eliminating the majority of my concerns. Based on the rebuttal, I changed my recommendation from "weak reject" to "weak accept" assuming that the authors will update their submission with the provided revision.

---

### Official Review · Reviewer_UhGW · 2023-07-18

**Confidence:** 1
**Originality:** Good
**Technical Quality:** Good
**Clarity Of Presentation:** Fair
**Impact:** 3

**Recommendation:**

Weak Accept: I recommend accepting the paper, but will not argue for my recommendation if the majority of other reviewers have a different opinion.

**Review:**

Strengths:
- the paper is the first to present the new task of multi-agent collaborative perception given both LiDAR and camera inputs
- the proposed method is reasonable and novel in its cooperate depth generation and bias multi-model fusion techniques
- the comparison to baselines is quite comprehensive and the proposed method achieves the best performance

Weaknesses:
- the paper writing can be improved, especially on the equations and figures. The figures are usually not well explained by the captions. Also, many operations in equations are not explained very clearly in the text.
- the novelty of the new problem is low as there are plenty works on studying how to combine lidar and camera inputs and also many works investigating multi-agent collaboration. The proposed problem is new, but not significantly different from other related tasks.

**Quality Of The Limitations Section:**

Limitations are addressed clearly

**Questions For Rebuttal:**

see weaknesses

**Robotics Focus:**

Relevant but unlikely to deploy to hardware in near future

**Summary Of Paper:**

This paper presents a novel problem formulation and a new framework for multi-agent collaborative perception given both LiDAR and camera inputs. Experiments are done using the DAIR-V2X dataset and the proposed method is experimentally shown to be superior than several baseline methods.

**Summary Of Recommendation:**

I have no experience in this field of study and my review should only be regarded as an educated guess. Overall, the proposed new task and method have merits, but maybe not very significant. The paper writing should be improved as well.

Post rebuttal: Thanks for authors' rebuttal. I've raised my score to Weak Accept.

---

### Official Review · Reviewer_bD4g · 2023-07-20

**Confidence:** 4
**Originality:** Good
**Technical Quality:** Good
**Clarity Of Presentation:** Good
**Impact:** 4

**Recommendation:**

Weak Accept: I recommend accepting the paper, but will not argue for my recommendation if the majority of other reviewers have a different opinion.

**Review:**

# Strengths:
* The problem this paper solving is interesting and important.
* Novel multi-modal feature fusion methods are proposed and evaluated in detail.
* Detailed illustration of method and experiment.
* Interesting analyze of the robustness against missing sensor

# Weaknesses:
* Analyze of the computational cost is missing


**Quality Of The Limitations Section:**

Limitations are addressed clearly

**Questions For Rebuttal:**

Some Camera-LiDAR related works could be included and compared in the "related works" section:

[1] "CamLiFlow: Bidirectional camera-LiDAR fusion for joint optical flow and scene flow estimation." Proceedings of the IEEE/CVF Conference on Computer Vision and Pattern Recognition. 2022.

[2]  "Perception-aware multi-sensor fusion for 3d lidar semantic segmentation." Proceedings of the IEEE/CVF International Conference on Computer Vision. 2021.

[3] "Advancing self-supervised monocular depth learning with sparse liDAR." Conference on Robot Learning. PMLR, 2022.

Line 174: LIDAR -> LiDAR

**Robotics Focus:**

Highly relevant to robotics but no hardware experiments

**Summary Of Paper:**

This paper proposed a multi agent collaborative perception framework with LiDAR-Camera modalities.
1, A LiDAR guided modal fusion strategy is proposed to fuse the LiDAR and camera feature.
2, Modality guided  collaborative fusion is proposed to achieve efficient and effective multi-agent communication.
3, The author reports superior performance on public datasets and robustness in handling missing camera sensor.

**Summary Of Recommendation:**

This paper has some novel and interesting ideas and is well written and presented.

---

### Decision · Program_Chairs · 2023-08-30

**Decision:**

Accept (Poster)

**Comment:**

This paper proposes an approach for collaborative perception with LiDAR and camera modalities. All the reviewers find the problem being addressed very interesting and the proposed solution is novel. In the initial review, the reviewers raised several concerns, most of which were addressed in the rebuttal. The major concern was the poor writing, concerns on generalization ability since the approach was only evaluated on one dataset, and clarity on the method requiring ablations. All the reviewers are satisfied with the responses and clarifications provided by the authors.